# Comparative Proteomic Analysis of Wild and Cultivated Amaranth Species Seeds by 2-DE and ESI-MS/MS

**DOI:** 10.3390/plants13192728

**Published:** 2024-09-29

**Authors:** Esaú Bojórquez-Velázquez, Jesus Alejandro Zamora-Briseño, Alberto Barrera-Pacheco, Eduardo Espitia-Rangel, Alfredo Herrera-Estrella, Ana Paulina Barba de la Rosa

**Affiliations:** 1Instituto Potosino de Investigación Científica y Tecnológica A. C., San Luis Potosí 78216, Mexico or esau.bojorquez@inecol.mx (E.B.-V.); alberto.barrera@ipicyt.edu.mx (A.B.-P.); 2Red de Estudios Moleculares Avanzados, Campus III, Instituto de Ecología A. C., Xalapa 91073, Mexico; alejandro.zamora@inecol.mx; 3Instituto Nacional de Investigaciones Forestales Agrícolas y Pecuarias, Texcoco 56250, Mexico; espitia.eduardo@inifap.gob.mx; 4Unidad de Genómica Avanzada, CINVESTAV, Irapuato 36821, Mexico; alfredo.herrera@cinvestav.mx

**Keywords:** *Amaranthus*, proteomics, two-dimensional gel electrophoresis, seed storage proteins, GBSSI

## Abstract

Amaranth is a promising staple food that produces seeds with excellent nutritional quality. Although cultivated species intended for grain production have interesting agronomic traits, relatively little is known about wild species, which can prosper in diverse environments and could be a rich genetic source for crop improvement. This work focuses on the proteomic comparison between the seeds of wild and cultivated amaranth species using polarity-based protein extraction and two-dimensional gel electrophoresis. Differentially accumulated proteins (DAPs) showed changes in granule-bound starch synthases and a wide range of 11S globulin isoforms. The electrophoretic profile of these proteins suggests that they may contain significant phosphorylation as post-translational modifications (PTMs), which were confirmed via immunodetection. These PTMs may impact the physicochemical functionality of storage proteins, with potential implications for seed agronomic traits and food system applications. Low-abundant DAPs with highly variable accumulation patterns are also discussed; these were involved in diverse molecular processes, such as genic regulation, lipid storage, and stress response.

## 1. Introduction

The combination of the steadily growing global population, estimated to reach around 9.3 billion by the year 2050, with the increasing difficulty of producing more crops in the face of climate change and soil deterioration has exacerbated the problem of meeting global food demand [1,2]. This has resulted in the need for the development of crops with improved agronomical features, such as better yields, resistance to biotic and abiotic stresses, higher nutritional quality, or faster growth rates [3,4,5].

In the quest for finding traits to improve grain-producing crops, orthodox seeds are irreplaceable key players as the central focus of crop production and food security since they act as safe containers of wide genetic information, which is preserved even under prolonged and unfavourable storage conditions [4,6,7,8]. In this sense, attempts to exploit the genetic potential contained in the seeds to develop crops better adapted to current and future challenges justifies efforts to collect, preserve, and deeply analyze the germplasm of commercial crop species as well as their wild relatives [2,4,7]. For this reason, it is particularly important to emphasize the significance of revaluing crops that have a high potential to aid in food security but are underestimated for various reasons. This is the case of *Amaranthus*, a genus that comprises both wild and cultivated species, the former used as vegetables and the latter cultivated for grain production, that has been used anciently as a staple food by pre-Hispanic civilizations, including the Aztec, Mayan, and Inca cultures. Unfortunately, its consumption has historically steadily dropped since its cultivation was prohibited due to its association with pagan rituals [9]. However, over the last three decades, research on amaranth’s nutritional and nutraceutical properties has revived its importance, opening new horizons in the history of amaranth cultivation [10,11]. At this time, the relevance of amaranth as a crop for human nutrition has been revalued, in part due to the high quality of its grain proteins.

Amaranth grain proteins have a high concentration of lysine and methionine, which are present in low levels in cereals and legumes, respectively. They also have almost no prolamin and contain nutraceutical compounds like bioactive peptides, lipids, phytochemicals, and carbohydrates [10,12,13,14]. Another interesting feature of amaranth plants is that they show resistance to different abiotic stresses such as salinity and drought and can be grown under rough conditions where other crops cannot [15,16]. Furthermore, there are significant potential genetic resources for their improvement, as only three amaranth species are destined for grain production (i.e., *A. caudatus*, *A. cruentus*, and *A. hypochondriacus*), whereas there are approximately 70 wild, unstudied amaranth species in nature that usually develop under a wide range of typically harsh environmental conditions [17,18].

Comparing the proteomic profiles of wild and cultivated species provides useful information about the proteins that contribute to key agronomical traits [19,20,21,22]. Proteomic analyses of crops such as soybean and jatropha have been useful for the identification of proteins involved with lipid content levels [23,24,25]. In barley, the use of comparative proteomics allowed the establishment of molecular markers for the best-performing cultivars in the malting process for beer production [26,27,28]. Its use also permitted the identification of cabbage and rice plants that are resistant to *Xanthomonas campestris* infection and the attack of the small brown planthopper, respectively [19,29].

Proteomic studies on amaranth have so far concentrated on two cultivated species: *A. hypochondriacus* and *A. cruentus* [16,30,31]. Only two publications from our group have addressed the comparison of protein profiles between wild and cultivated amaranth species, using one-dimensional sodium dodecyl sulphate-polyacrylamide gel electrophoresis (1D-SDS-PAGE) for the analysis of both total protein extracts and polarity-based protein fractionations [32,33]. Although these studies allowed us to identify the fundamental differences at the proteomic level amongst seeds of amaranth species, more research is needed to detail these findings and explain differential physiological characteristics previously observed, such as grain size and form, seed coat (testa) thickness, lipid profile, and starch composition and packaging. Therefore, here we analyzed the amaranth seed proteomes via two-dimensional gel electrophoresis (2-DE) to broaden the search for seed nutritional quality and physiology-relevant proteoforms of wild and cultivated amaranth species.

## 2. Results and Discussion

### 2.1. Establishment of 2-DE Proteomic Maps by Polarity-Based Extraction and Global Protein Identification

The high-resolution 2-DE proteomic maps for the hydrophilic (H1) and hydrophobic (H0) fractions of each amaranth species are presented in Figure 1 (Appendix A).

The H1 fraction was characterized by the presence of protein spots in a wide range of MWs, from less than 10 kDa to higher of 130 kDa, with the most abundant spots at the low (10–15 kDa) and medium (25–40 kDa) MW intervals. In this fraction, 152 spots were detected as DAPs amongst the amaranth species, from which 99 were successfully identified via nLC-MS/MS, for a total of 59 unique proteins (Appendix A). The DAP patterns in the H1 fraction were noticeable in the low MW region, nearly to 10 kDa, where it could be observed how some spots only belong to certain species (Figure 1), reflected mainly by the spots H1-25, H1-31, H1-614, H1-638, H1-652, H1-653, H1-656, H1-657, H1-659, H1-667, H1-670, and H1-689. Of these, only spots 31 and 670 were identified: Vicilin-like seed storage protein (SSP) and seed maturation protein (SMP), respectively. The former accumulates mainly in cultivated species and the second is present in both *A. hypochondriacus* cultivars and the wild species *A. hybridus*.

The proteomic map analysis of the H0 fraction revealed 120 spots detected as DAPs, of which 92 were identified, giving 35 unique proteins (Appendix A). The H0 fraction showed two groups of markedly intense spots; the first was located at 35 kDa, and the second, denominated as the High Variation Region (HVR), was located in the range of 55–70 kDa, distributed in the form of three series of “beads on a necklace” or “trains” located in the upper part of the alkaline region of the proteomic maps. The H0 fraction was characterized by the identification of three main proteins broadly distributed in several spots: 11S globulin (021282) in 38 spots, granule-bound starch synthase I (GBSSI, 011500) in 30 spots, and Vicilin-like SSP (018839) in 13 spots.

The comparative proteomics approach using 2-DE has been widely used to evaluate different varieties and plant species, both model, such as *Arabidopsis thaliana* [34], and crops of agronomic and economic relevance, for example, barley [26,35], rice [36] wheat [37,38], and soybean [25,39]. Although there are reports about the proteome of amaranth seed [30,31], these are focused on a descriptive outlook using total protein extracts of only one cultivar of *A. cruentus*, and until now, no works have compared the proteomes of wild and cultivated species using differential extraction approaches and 2-DE. The main advantage of the polarity-based approach used in this work is that it allowed for the extraction of low-abundant proteins in the hydrophilic fraction and high-abundant proteins in the hydrophobic fraction, with the latter mainly constituted of GBSSI that is tightly bound to starch granules and is the most abundant protein encapsulated within starch, and by 11S globulin-type SSPs, which have been classified as glutelins with a high degree of aggregation probably due to the formation of multiple intermolecular disulfide bonds. It is for these reasons that GBSSI and some isoforms of amaranth 11S globulins can be only extracted and solubilized in the presence of denaturants, reducing agents, and chaotropic agents [40,41]. The amaranth seed proteomic maps of the H1 and H0 fractions are highly similar to those of the whole seed and perisperm, respectively, reported for sugar beet (*Beta vulgaris*) seeds [42]. Our approach allowed us to overcome the dynamic range dilemma without the need to carry out tissue dissections or long extraction processes that require multiple time-consuming steps, such as the method established by Osborne, which is widely used in the study of seed proteins [43,44].

### 2.2. Grouping of Amaranth Species Based on the Abundance of Protein Fractions

The abundances of differential spots of the H1 and H0 fractions were analyzed as one single data set, leading to a clear separation between the samples, as shown in the PCA presented in Figure 2. Amaranths can be readily classified into distinct groups. Wild species are clustered apart from cultivated ones, with *A. hypochondriacus* cultivars positioned closer together while being separated from *A. cruentus.* The correlogram and heat map shown in Figure 3 reveal the contributions of the total differential spot abundance and agglomerative hierarchical clustering in concordance with the PCA. Similarly, when examining only the spots in the H1 proteomic maps, *A. hypochondriacus* cultivars and wild amaranths are found in the same clade but in separate groups, while *A. cruentus* is not grouped with the other species.

Most of the spots in the H1 fraction that support species differentiation, as shown in Figure 1, were found to be isoforms of the SSPs vicilins (7S globulins) and legumins (11S globulins). In general, these proteins presented significantly higher abundance in the wild species *A. hybridus* and in the *A. hypochondriacus* cultivars. Spots H1-151 and H1-157, identified as 11S globulin (001411), had higher levels of accumulation in the two *A. hypochondriacus* cultivars. Spots H1-218 and H1-362, identified as vicilin-like SSP (018839), were predominant in *A. hypochondriacus* cv. Cristalina, whereas spot H1-821, vicilin-like SSP At2g18540 (010140), was mainly present in *A. hybridus*. Vicilin-like (018839) and 11S globulin (021282) were identified in spots H1-213 and H1-370; both proteins were present in spot H1-370, but in spot H1-213, only Vicilin-like was found. The highest abundance on these spots is shared between *A. hybridus* and both *A. hypochondriacus* cultivars.

In contrast, when only differential H0 spots abundances are considered, the aggrupation of amaranth species changes (Figure 3): *A. hypochondriacus* cv. Cristalina is clustered with the wild amaranths, whereas *A. hypochondriacus* cv. Opaca and *A. cruentus* are grouped together, forming a separated clade. This is because *A. hypochondriacus* cv. Cristalina is the unique cultivable species that shares the same sets of spots trains in the HVR with the wild species and represents a considerable number of spots in which 11S globulin (021282) and GBSSI (011500) were identified. GBSSI was found in spots H0-9, H0-11, H0-12, H0-13, H0-15, H0-16, H0-20, H0-25, H0-566, and H0-568, located in the upper string of the HVR. 11S globulin was found in spots H0-518, H0-527, H0-533, H0-536, H0-537, H0-541, and H0-543, corresponding to the lower string.

### 2.3. The Diversity of GBSSI and 11S Globulins Proteoforms Amongst Amaranth Species Could Be Explained by the Presence of Post-Translational Modifications (PTMs)

GBSSI and SSPs were the most prevalent proteins in the spots that showed differential accumulation between amaranth species in both the H1 and H0 fractions. They also contributed most strongly to the segregation of the species. GBSSI was identified in three spots of H1, and its accumulation patterns present a complementary scheme between species (Figure 4). Spot H1-403 was over-accumulated in all species except *A. hybridus*, while spot H1-426 presented an opposite trend; spot H1-427 remained homogeneous but slightly increased in *A. hybridus*. Regarding the H0 fraction, the two wild species and *A. hypochondriacus* cv. Cristalina were grouped in the same clade, whereas *A. cruentus* and *A. hypochondriacus* cv. Opaca were placed in different clades.

Concerning SSPs, the heatmaps shown in Figure 5 depict the accumulation data of spots where 7S and 11S globulins were identified as the main proteins based on emPAI. In the H1 fraction SSPs, *A. cruentus* stands apart from other species due to the higher accumulation of spots H1-231, H1-682, and H1-755, which were identified as Vicilin-like SSP (018839). The spots with the highest abundance in the five species for this fraction were H1-157, H1-213, and H1-821, which were identified as 11S globulin (001411), Vicilin-like SSP (018839), and Vicilin-like SSP At2g18540 (010140), respectively. In H0, the SSPs classified the wild species *A. hybridus* and *A. hypochondriacus* cv. Cristalina as belonging to the same clade since they share the overaccumulation of 11S globulin (021282) in spots H0-103, H0-536, and H0-541. The accumulation of spot H0-533, with high abundance in all species, stands out, but it is outstanding in *A. powellii*, *A. cruentus*, and *A. hypochondriacus* cv. Opaca, which also corresponds to 11S globulin (021282) and is present in the HVR.

Due to the wide diversity of spots in which both GBSSI and 11S globulins were identified, mainly in HRV of H0, and the reduced number of accessions obtained, that is, unique proteins, we suggest that the change in *pI* values is a result of the presence of a significant number of post-translational modifications (PTMs), with phosphorylation being one of the main PTMs that affect the net charge of polypeptide chains. Based on this, the immunodetection of phosphorylated proteins was carried out using anti-phosphoserine/threonine/tyrosine polyclonal antibodies, and the result is shown in Figure 6. The electrophoretic profile showed three bands of higher intensity, corresponding to the HVR formed by three sets of “beads on a necklace” that differ in MW observed by 2-DE in the range of 55–70 kDa. There were also high-intensity bands in the 35 kDa and 20 kDa regions that belong to the acidic and basic subunits, respectively, of the 11S globulins after the cleavage of their interchain disulfide bond. The signals in the Western blot were resolved mostly in the areas corresponding to these bands, indicating the presence of phosphorylation in GBSSI and 11S globulins from amaranth seeds.

As mentioned above, the HVR is formed by three sets of “beads on a necklace”: the largest (70 kDa) is present in *A. hybridus*, *A. powellii*, and *A. hypochondriacus* cv. Cristalina; the intermediate (65 kDa) is present in *A. powellii* and *A. cruentus*; and the lowest (55 kDa) was observed in *A. hybridus* and in the two varieties of *A. hypochondriacus*. The 70 kDa band corresponds to the functional GBSSI (011500). In contrast, in the 65 kDa and 55 kDa bands, the predominant protein was the 11S HMW globulin (021282) previously reported [32]. The immunodetection signal matches the Coomassie blue staining for the three bands of the HVR but with greater intensity for those of 65 and 55 kDa. Interestingly, the 55 kDa band observed via immunodetection for *A. powellii* and *A. cruentus* was not detected via Coomassie staining. With the advent of second-generation shotgun proteomics techniques, massive studies based on 2-DE have decreased; however, electrophoresis-based approaches present advantages in certain applications, such as the visualization of isoforms and the detection of PTMs [45,46]. The identification of proteo- or iso-forms in the HVR of H0 proteomic maps is a clear example of this. The HVR amaranth SSPs display an interesting pattern and higher intensity together with GBSSI. These “beads on a necklace” or “trains” shaped spots in 2-DE profiles are characteristic of changes in *pI* due to PTMs, mainly phosphorylation [45,47,48].

Phosphorylation and other charge-modifying PTMs, like methylation or acetylation, can play important roles in the regulation of the activity of the proteins in question; for example, in the case of GBSSI, the degree of phosphorylation would be a mechanism to modulate the rate of amylose synthesis. Regarding 11S globulins, this modification may be involved in the stabilization or breakdown of the quaternary structure of the protein, or function as a signalling mechanism for transport and mobilization during germination when stored nutrients are required [47,49,50,51].

GBSSI is responsible for amylose biosynthesis and is localized exclusively within the starch granule; it is also involved in the elongation of long chains in amylopectin [52]. Amaranth seeds with functional GBSSI synthesize structured and translucent amylose-containing perisperms, i.e., non-waxy phenotype, as in the case of wild species *A. hybridus* and *A. powellii*, and the cultivated *A. hypochondriacus* cv. Cristalina [32,33]. Species or cultivars that lack this protein or that synthesize an incomplete non-functional version of GBSSI produce starches consisting only of amylopectin, which gives rise to amorphous or agglutinated perisperms, which is known as the waxy phenotype [53].

GBSSI has been reported to be a phosphoprotein in several phospho-proteomic studies, mainly focusing on rice, maize, and wheat, and has been identified in up to 12 spots within the same 2-DE gel, forming “beads on a necklace” patterns, as was observed here for amaranth seeds [51,52,54,55,56]. In this protein, phosphorylation is a dynamic process during the different stages of seed development; it has nine sites that are susceptible to this modification; therefore, the degree of phosphorylation has been suggested to be a mechanism for regulating the activity of this enzyme in a species-specific way. In wheat, at least 10 spot isoforms for GBSSI were reported, whose accumulation varies in time from 10 days postanthesis to the mature stage [51,54]. In rice GBSSI, a phosphorylation/oligomerization-dependent activity-regulating mechanism has been described. Throughout this process, the degree of protein phosphorylation controls oligomer formation since there is a transition balance between monomeric and oligomeric species during the development of the endosperm, and it has been observed that the enzymatic activity is increased in the oligomeric form, whose formation is favoured at higher levels of phosphorylation [57].

SSPs have been reported to be the main variable proteins in the comparison of different plant cultivars. For amaranth species, they account for 38% of the variations in the differential proteins, which are given by only three proteins: 11S globulin, Vicilin-like, and Vicilin-like At2g18540. These behaviours have been observed in soybean and wheat, where SSPs represent up to 54% and 40% of the variations, respectively [37,39]. There is a heterogeneous distribution of SSPs. The same protein is more abundant in different spots; for example, in the hydrophobic fraction, 11S globulin is characteristic of *A. hybridus* in spots H0-103, H0-494 and H0-536; *A. powellii* is characteristic in spot H0-544; and *A. cruentus* is characteristic in spots H0-405, H0-460, H0-477, H0-481, and H0-501. This behaviour has been observed in *A. thaliana* and soybean [39,49]. This performance is also reflected in the case of other proteins like seed biotin-containing protein SBP65 (013747), which is increased in wild species in spots H1-466 and H1-467 but has higher levels in *A. hypochondriacus* cultivars (spot H1-488)], OBAP 1A, agglutinin, and MetE. This may be because homologous versions of the proteins for each species show variations in their amino acid sequences, impacting *pI* and/or MW and placing them in different positions on the proteomic maps.

Another alternative may be the presence of PTMs or a discrepancy in terms of proteolytic processing, which has been widely described for SSPs [58,59]. In *A. thaliana*, four cruciferin proteins (CruA, CruB, CruC, and CruD; 11S globulins belonging to the Brassicaceae/Cruciferae family) encoded by a different gene each were immunodetected using antibodies against phosphorylated amino acid residues (pSer, pThr, and pTyr) and established as the major phosphorylated proteins in *A. thaliana* seeds. These findings were corroborated using LC–MS/MS analysis of cruciferins extracted from the *abi1-1* mutant, which lacks the corresponding encoding protein phosphatase, in the absence of which, the level of phosphorylated sites and proteins increases. This allowed for the identification of a total of 20 phosphorylation sites; eight were located in the α-subunits and eleven were located in the β-subunits [49]. Native 11S globulins are hexamers where two trimers interact face-to-face; therefore, this area of interaction is not exposed to the surrounding solvent or environment; however, studies using molecular modelling and dynamics showed that most of the potential phosphorylation sites in CruA or CruB are located at the trimer interface, which inaccessible to modifications [60]. However, once the hexamer dissociates into two trimers after imbibition during germination, these hidden sites can be exposed to the surrounding solvent and phosphorylated for signalling and transportation purposes. Here, the amaranth 11S globulin was identified as a phosphoprotein by using immunodetection; however, phosphopeptide enrichment is required in future works to identify the specific sites of this modification and investigate its possible biological and biotechnological repercussions.

### 2.4. Differentially Accumulated Non-Abundant Proteins Related to Metabolic Processes

In addition to the main differences relating to protein accumulation amongst amaranth seed species represented by GBSSI and SSPs, the identification results also showed a broad set of low-abundance proteins involved in various metabolic processes, such as gene regulation, responses to reactive oxygen species, the storage of lipids, stress responses, and central and amino acid metabolism (Appendix A).

#### 2.4.1. Genic Regulation

Orthodox seeds are quiescent, and their genomes are subjected to epigenetic modifications that maintain their inactive status, preventing premature transcription and seed germination. It has been shown that histone modifications play a role in chromatin packing and accessibility, nucleosome mobility, gene expression, and chromosome arrangement [61]. Histones are highly heterogeneous chromatin proteins with a *pI* of ~11 and an MW of 10–22 kDa. Histone proteins can undergo multiple PTMs at their amino terminal “tails”, including acetylation, methylation, phosphorylation, ubiquitination, and ribosylation, which alter the structure of chromatin and its function [62,63]. The acetylation of histones represents an epigenetic mechanism with a role in the transcriptional control maintenance of gene expression patterns and cell memory [64]. Only mono-acetylated isoforms of H4 were found in dry seeds, and increased acetylation is typically associated with gene activation due to a more open chromatin structure that makes it accessible to transcription factors [65]. It would be expected that if a change in one of the histones is observed, a variation of the same magnitude should be reflected in the rest of them, as reported for their upregulated transcript levels under drought conditions [66]. However, only one of the four core histones, histone H4, was detected as being differentially accumulated, with higher abundance in *A. powellii* (spot H1-28). Histones are small and highly basic proteins with *pI*s of 10–12; therefore, they should be left out of the 2-DE analysis since IPG strips with a range of 5–8 were used in this work; however, the experimental *pI* observed here for histone H4 was 5.1, which is probably due to the presence of PTMs that impact this property. Despite this technical limitation, it should be noted that only histone H4 from *A. powellii* seeds could have a PTM profile that confers an acidic *pI* to this protein and is possibly related to a specific form of epigenetic regulation.

#### 2.4.2. Reactive Oxygen Species (ROS) Scavenging

DNA damage is one of the major cellular defects associated with seed deterioration. Even in the quiescent state of the desiccated seed, DNA damage is provoked by the formation of reactive oxygen species (ROS). During seed storage, DNA repair is hampered by the low water content. Therefore, DNA lesions can accumulate, and DNA repair should be initiated immediately upon imbibition. Poly(ADP-ribose) polymerases (PARPs) are important components in DNA damage response. In *A. thaliana*, AtPARP3 was very highly expressed in seeds, while null mutants in this gene displayed a delay in germination, indicating that AtPARP3 is an important component of seed storability and viability [67]. In this work, one isoform of PARP3 (spot H1-492) was detected in high abundance in wild species, followed by *A. hypochondriacus*, while the lowest abundance was detected in *A. cruentus*. A second PARP3 isoform (spot H1-497) showed a profile of increasing abundance from wild to cultivated *A. hypochondriacus* species. The third isoform (spot H1-518) showed higher abundance in *A. hypochondriacus* cultivars. This could indicate that *A. hypochondriacus*, as a cultivable species, is more prone to DNA repair after water imbibition and higher germination rates.

Seed viability and longevity also rely on antioxidants and ROS scavengers [1]. ROS are metabolic byproducts of oxygen reactions, and because of its reactivity, oxygen concentrations should be kept below toxic levels. However, ROS also act as signalling molecules that are implicated in development and stress response gene expression regulation [68]. Antioxidant and redox-related enzymes, such as gluthathione-S-transferase and superoxide dismutase (SOD), were detected in amaranth seeds. Glutathione-S-transferase (DHAR) was detected in lower abundance in *A. cruentus* (spot H1-53). DHAR or dehydroascorbate reductases are active thiol transferases that are known to be antioxidant enzymes but lack glutathione conjugating activity [69]. SODs are one of the most effective defence systems in plant cells against ROS toxicity. In *Ricinus communis*, SOD genes were related to greater stress responses [70]. They are classified according to the metal cofactor that they bind to (e.g., Cu/Zn-SOD, Mn-SOD, Fe-SOD, and Ni-SOD) and their sub-cellular location [71]. Fe-SOD, which is localized in plastids, was identified as being more abundant in *A. hypochondriacus* cv. Cristalina (spot H1-483).

CBS domain-containing proteins directly regulate the activation of thioredoxins and thereby control cellular H_2_O_2_ levels and modulate both plant development and growth [72]. The knockdown of CBSX3 in *A. thaliana* revealed anther indehiscence due to deficient lignin deposition caused by insufficient ROS accumulation, as well as an increased expression of genes related to cell cycle and accelerated plant growth. However, in CBSX3-overexpressing plants, ROS accumulation increased, and cell cycle-related gene expression decreased; thus, plant growth was retarded and leaf size decreased. Then, it was suggested that CBSX3 activates o-type thioredoxin (Trx-o2) in mitochondria in plants and plays an important role in regulating plant development and the redox system [73]. CBS domain-containing protein mitochondrial (CBSX3, spot H1-767) was found to be upregulated in *A. cruentus*.

Seed dormancy is mainly controlled by abscisic acid (ABA) and gibberellin (GA) and can be classified as primary or secondary seed dormancy. Primary seed dormancy is induced by maternal ABA. In *A. thaliana*, AtPER1 is a seed-specific peroxiredoxin involved in enhancing primary seed dormancy [74]; furthermore, this gene can also eliminate ROS by suppressing ABA catabolism and GA biosynthesis, improving primary seed dormancy and making the seeds less sensitive to adverse environmental conditions [75]. A Peroxiredoxin-2B (spot H1-760) was detected at a low abundance in *A. powellii* and *A. cruentus*.

#### 2.4.3. Lipid Storage

Storage lipids in seeds are predominantly accumulated in specialized cytoplasm organelles termed lipid bodies, oil bodies, or oleosomes. Oil bodies (OBs) are also used as transient storage depots for proteins that lack appropriate binding partners in the cell and as a general cellular strategy for handling excess proteins [76]. They seem to be involved in protecting plant embryos against freezing. Several oil body-associated proteins (OBAPs) were identified in the amaranth species. In the H1 fraction, OBAPs had a significantly higher abundance in *A. cruentus* and *A. powellii*, but as different homologues, OBAP 1A (009953, spot H1-112) had higher levels in *A. powellii*, and OBAP 2A (004342, spot H1-106) was more accumulated in *A. cruentus*. In the H0 fraction, OBAP 1A displayed variable profiles; it was more accumulated in *A. hybridus* in spot H0-353; it had a higher abundance in *A. powellii* and *A. cruentus* in spot H0-354 as well as in *A. hybridus* and *A. cruentus* in spot H0-355; and greater levels were also found in both *A. hybridus* and *A. hypochondriacus* cultivars in spot H0-357. It has been shown that the expression of OBAP1 decreases rapidly after germination, while *obap1a* mutants decrease in terms of germination rate. Furthermore, there is a decrease in seed oil content, changes in fatty acid composition, and their embryos have few, big, and irregular oil bodies compared with the wild type [77]. Previous works have shown that *A. powellii* is the species with the highest fat content and *A. hybridus* have the lowest fat content amongst the species analyzed. It has also been shown that the compositions of seed oils are different between amaranth species and cultivars, and due to this, each species or variety of amaranth may adjust the quantity and type of OBAPs that are required to stabilize the packaging of these hydrophobic biomolecules [33].

Peroxygenases are a group of heme-containing monooxygenases that catalyze the hydroperoxide-dependent epoxidation of unsaturated fatty acids. These enzymes can catalyze the hydroxylation of aromatics, sulfoxidation of xenobiotics, and, most frequently, the epoxidation of unsaturated fatty acids. They belong to the caleosin family, a group of oil-body-associated proteins [78]. Two isoforms of peroxygenase were detected in amaranth seed proteomes; the first was found in higher abundance in *A. hybridus* and *A. cruentus* (spot H0-348), and the second was found in *A. hybridus* (spot H0-365).

#### 2.4.4. Stress Response

Crop yields are often affected by different kinds of biotic and abiotic stresses. Heat shock proteins (HSPs), late embryogenesis abundant proteins (LEAs), embryonic protein DC-8, and annexin, which were identified amongst the DAPs in amaranth species, have been reported to be involved in responses to different types of stress.

HSPs are ubiquitous proteins classified according to their size; they act as molecular chaperones, thus favouring protein folding and preventing protein aggregation [68]. In amaranth seeds, four HSPs were identified: 17.4 kDa class I (spot H0-248), 17.9 kDa class II (H0-197), 18.3 kDa class I (spots H1-72, H1-771, H1-772, and H0-233), and HSP 70 kDa (spot H1-449). All of them were significatively more abundant, almost only in *A. cruentus*, but HSP 70 kDa was also highly abundant in *A. powellii*. It is worth highlighting the fact that 18.3 kDa class I HSP was identified in three different isoforms in the H1 fraction and one in the H0 fraction, which vary in terms of MW and *pI*. This is indicative of processing and/or PTMs, which reflects that different proteoforms of this HSP are present in the mature seeds. HSPs have been reported during seed development in common bean and wheat varieties [79,80]. During the early stages of wheat grain development, HSP70 levels increase; conversely, in common beans, HSP83 decreases between 10 and 20 days postanthesis and remains stable after that. High levels of these proteins have been related to increased vigour and better germination rates [81]. Another protein identified in this category was ClpB, which belongs to the casein lytic proteinase/heat shock protein 100 (Clp/Hsp100) family [82]. Seeds of many plant species, such as wheat, maize, and mustard, contain high levels of ClpB protein located in chloroplasts. These levels have been linked to plant protection against heat stress [83]. ClpB1 (spot H1-516) was more abundant in *A. powellii* and less abundant in *A. hypochondriacus* cv. Cristalina.

LEA proteins play important roles in resistance to abiotic stress, mainly related to desiccation and water deprivation stress [84]. Two LEA31 isoforms were detected in the H1 fraction; spot H1-103 had a significantly higher abundance and spot H1-130 showed a significantly lower abundance in *A. powellii*. In *A. thaliana*, LEA31 is encoded by the gene At3g22490, also known as Atrab28 (responsive to abscisic acid 28), the overexpression of which confers faster germination under control and osmotic stress conditions [85]. Like the ortholog of *A. thaliana*, amaranth LEA31 has three seed maturation protein motifs, implying that it has a similar role during seed development and germination. In the H0 fraction, LEA B19.3 was accumulated at high levels in *A. cruentus* (spot H0-168); this protein shares a strong similarity to the highly hydrophilic Em protein. The physicochemical characteristics of LEA B19.3 contrast with the hydrophobic fraction in which it was extracted; therefore, it is possible that during the obtention of the H1 fraction, it remained non-covalently linked to its molecular target due to it being insoluble and was released with denaturalizing and chaotropic agents. B19 genes are similarly regulated during embryo development, but a different form of regulation was observed in relation to ABA, osmotic stress, and salt stress [86].

Embryonic protein DC-8 was identified in two spots, H1-454 and H1-457; the first was significatively lower while the second was higher in *A. cruentus*. DC-8 is one of a group of genes that appear to be regulated by ABA, and their corresponding proteins may function during the desiccation and maturation phase of embryogenesis. In carrot, DC-8 encodes a hydrophilic 66 kDa protein located in the cytoplasm and cell walls of the embryo and endosperm [87]. The experimental masses of the amaranth isoforms are of 79.0 and 85.7 kDa; thus, they could be unprocessed or immature versions of the protein.

SMP (spot H1-670) was significatively higher in both cultivars of *A. hypochondriacus*. The SMP domain of the AfrLEA6 protein contributes to desiccation tolerance in *Artemia franciscana* by increasing cytoplasmic viscosity and providing protective compartments for desiccation-sensitive proteins [88].

Annexins are cytosolic proteins involved in signal transduction pathways that can attach or insert into plasma membrane or endomembrane depending on the cytosolic free Ca^2+^ concentration, pH, lipid composition, or membrane voltage. In plants, these proteins have been found in almost all types of tissues, including seeds, roots, stems, and leaves, and are implicated in a wide range of processes, such as exocytosis, cell elongation, wall synthesis, nodulation, and fruit ripening, but they are mainly associated with Ca^2+^ and reactive oxygen species homeostasis due to their actions involved in against abiotic stress conditions, for example, cold, oxidative, saline, and abscisic acid responses [89,90]. The annexins identified here have greater levels of accumulation in both wild amaranth species and were identified in the hydrophobic fraction (spot H0-404). This implies that, in seeds, this protein is linked to membranes. In seven-day-old plants, the transcript level of the *A. thaliana* orthologue (AnnAt8) was reported to have a notable increase in response to dehydration; these stress conditions are analogous to those of seeds due to their low water content; thus, amaranth annexin might perform some protective role until germination begins [91].

#### 2.4.5. Central Metabolism Related Proteins (Glycolysis-TCA)

Orthodox seeds maintain low metabolic activity; however, the coexistence of anaerobic as well as aerobic energy metabolism in seeds is supported by the presence of proteins corresponding to enzymes involved in glycolysis and the TCA cycle [92]. This can explain why only a few enzymes related to carbohydrate, lipid, and nitrogen metabolism were identified as DAPs in amaranth seeds. Belonging to the glycolytic pathway, two isoforms of enolase were detected; the first (spot H1-348) was found to have significatively higher abundance in *A. powellii*, while it was significantly lower in *A. cruentus*; the second isoform was identified in spot H0-547, with high abundance in *A. cruentus*. Enolase is a key glycolytic enzyme that is essential for the growth and development of plants. Studies have demonstrated that the AtENO2 mutation in *A. thaliana* reduces the size and weight of seeds, leading to high levels of glucose and starch content [93]. Higher levels of enolase have also been related to cold tolerance and high-dormancy seeds [94,95].

In *A. powellii*, the highly abundant spot of H1-425 was identified as being the enzyme 2,3-bisphophoglycerate-independent phosphoglycerate mutase (iPGAM). Reports indicate that iPGAM is important in plant development, with a pivotal role in 3-phosphoglyceric acid (3-PGA) metabolism [96]. Glyceraldehyde 3-phosphapte dehydrogenase (G3P-DH) was identified as having a higher abundance in *A. hybridus* (spot H0-443) and *A. cruentus* (spot H0-446).

Fructose 1,6 bisphosphate aldolase (FBA) had a significantly higher abundance in *A. hybridus* (spot H0-451). FBA is a key enzyme in plants, and it is involved in glycolysis, gluconeogenesis, and the Calvin cycle. This enzyme can bind several partner proteins that impact cellular scaffolding, signalling, transcription, and motility [97]. FBA also plays significant roles in biotic and abiotic stress responses and in growth and development regulation [98].

The enzymes related to the TCA cycle identified in amaranth seed included two isoforms of malate dehydrogenase (spots H1-194 and H1-196); the first was significantly lower in *A. cruentus*, and the second was highly abundant in *A. powellii*. Both spots were found to have lower *pI* when compared with the theoretical data. Isocitrate dehydrogenase (spot H1-292) showed a lower abundance in *A. cruentus* and a higher abundance in wild species; this protein showed a higher *pI* than that in the theoretical data. The increased activity of NADPH-dependent dehydrogenases, such as isocitrate dehydrogenase and malic enzymes, has been reported under oxidative and salinity stress [35,99]. Lysine acetylation (Lys-Ac) is one of the major PTMs involved in plant responses to environmental signals. It can change the charge state of a protein, and therefore, its *pI*. Site-directed mutagenesis enzymatic assays showed that Lys-Ac strongly modified the activities of two key enzymes of primary metabolism: pyruvate dehydrogenase and isocitrate dehydrogenase [100]. Then, it was proposed that Lys-AC could be an important strategy for reconfiguration of metabolic processes during bud dormancy release [101]. Thus, the variations between the experimental and theoretical *pI*s observed for malate dehydrogenase and isocitrate dehydrogenase could be due to Lys-Ac.

NADP-dependent malic enzyme (spot H1-436) was found to have a higher abundance in *A. powellii*, followed by *A. cruentus*. Malic enzyme is an important enzyme in the photosynthesis of C4 plants; it catalyzes the decarboxylation of malate, generating pyruvate, CO_2_, and NADH or NADPH. In castor seeds, NADP-ME expression levels are high when lipid deposition is active [102].

The presence of glucose and ribitol dehydrogenase (Glc/RibDH) was detected as having a high abundance in wild species, with equal levels of accumulation in cultivated species (spot H1-145). Glc/RibDH catalyzes the oxidation of D-glucose (but no sugar phosphates) using NAD+ as a co-substrate. It has been shown that salinity-tolerant barley lines expressed a higher level of 6-phosphogluconate dehydrogenase and glucose/ribitol dehydrogenase (Glc/RibDH) [35].

#### 2.4.6. Cell Wall-Related Enzymes

The bifunctional UDP-glucose 4-epimerase and UDP-xylose 4-epimerase1 were highly abundant in *A. hybridus* (spot H0-444). UDP sugars serve as substrates in the synthesis of cell wall polysaccharides. The flux through nucleotide sugar interconversion reactions is one of the main steps controlling the amount of cell wall polysaccharides in plants. Nucleotide sugars are principally generated through *de novo* pathways, in which they are formed from a starting substrate, such as UDP-Glc or GDP-Man. The enzyme α-xylosidase 1 was detected in spot H0-72 at a higher abundance in *A. hybridus*, followed by cultivated species; this protein has cell wall and growth-modulating functions, contributing to the maintenance of the mechanical integrity of the primary cell wall. The overaccumulation of α-xylosidase causes a size reduction in the xyloglucan chain in growing tissues and germinating seeds, enhancing cell wall loosening [103]. UDP-D-apiose/UDP-D-xylose synthase 2 (spot H1-301) was significantly higher in *A. powellii*; this enzyme is related to cell wall formation, and its absence results in cell wall thickening and cell death [104].

#### 2.4.7. Amino Acid Metabolism

Regarding amino acid metabolism, three proteoforms of MetE were found distributed in seven different spots. Spots H1-444 (017360) and H1-472 (022179, 017357, 017360) had lower abundance only in *A. cruentus*; spots H1-478 (022179, 017,357 and 017360), H1-479 and H1-480 (017357, 017360) had higher levels in *A. powellii*; spot H1-494 (017360) was more accumulated in *A. hypochondriacus* cv. Cristalina; and spot H1-503 (017357, 017360) had greater accumulation in *A. hybridus*. Spot H1-444 was the only one corresponding to the experimental and theoretical MW/*pI* values; all other spots showed higher MW or even lower *pI* (spot H1-494).

In plants, the *de novo* synthesis of Met from the precursor O-phosphohomoserine requires only three enzymes: cystathionine γ-synthase, cystathione β-lyase, and MetE (also known as cobalamin-independent methioninesynthase) [105,106]. Reports have suggested that cobalamin-independent methionine synthase was decreased and increased in drought-sensitive and tolerant genotypes, respectively, suggesting that this enzyme is involved in the tolerance of fennel leaf under drought stress [107].

Methionine has an essential role in cellular metabolism as it is necessary for protein biosynthesis and is the precursor for the biosynthesis of S-adenosylmethionine (SAM), polyamines, and the phytohormone ethylene. SAM is a substrate of several reactions driving the biosynthesis of lignin [108]. S-adenosylmethionine synthase 4 was found in spot H0-491, with a higher accumulation being found in *A. hybridus*.

Ω-amidase (spot H1-202) was more abundant in wild species and had a very low abundance in *A. cruentus*. In Arabidopsis, ω-amidase may act as a critical enzyme in asparagine transamination. Asparagine is involved in inorganic ammonium assimilation utilization, nitrogen recycling, storage, and transport in response to environmental and internal signals, as well in primary processes, such as seed germination, vegetative growth, flowering and senescence, and seed filling and maturation [109]. Ω-amidase was used to increase nitrogen use efficiency, resulting in enhanced growth, faster growth rates, a greater number of seeds and fruit/pods yields, earlier and more productive flowering, increased tolerance to high salt conditions, and increased biomass yields [110].

## 3. Materials and Methods

### 3.1. Biological Material

Seeds of two wild amaranth species, *A. hybridus* (Ahy) and *A. powellii* (Apow), and three varieties of cultivated amaranth species, *A. cruentus* cv. Amaranteca (Acru), *A. hypochondriacus* cv. Opaca (AhyOp), and *A. hypochondriacus* cv. Cristalina (AhyCris), were kindly provided by the National Institute for Forestry, Agriculture, and Livestock Research (INIFAP), Mexico. The seeds were germinated on the BM2 Germination Mix sterile wet substrate (Berger, CA, USA) and placed at 4 °C for 12 h and transferred to a growth chamber at 25 °C and subjected to cycles of 12 h light/dark. After 11 days in the growth chamber, the seedlings were transferred to a greenhouse in plastic pots of 10 cm in diameter and 20 cm in height containing the same substrate. The plants were watered every third day until the reproductive stage concluded. Four biological replicates were harvested, which were composed of the seeds of five plants each. Protein extraction was carried out independently of each biological replicate.

### 3.2. Protein Extraction

Amaranth seed proteins were extracted using a sequential polarity-based method described by Bojórquez-Velázquez et al. (2019) [32]. Briefly, 500 mg of defatted seed flour was mixed using a vortex at a 1:20 (*w*:*v*, sample/buffer) ratio with a buffer composed of 10% glycerol (*v*/*v*) (Sigma-Aldrich, St. Louis, MO, USA) and 0.1 M 2-Amino-2-hydroxymethyl-propane-1,3-diol (Tris-HCl) (Sigma-Aldrich) with pH 8.0 for 15 min at 4 °C. Then, it was centrifuged at 17,000× *g* for 30 min at 4 °C in a Beckman Avanti J-26S XPI centrifuge (Beckman Coulter, Brea, CA, USA), and the supernatant was recovered and named the hydrophilic fraction (H1 fraction). The resulting pellet was resuspended and mixed using a vortex with a denaturing solution [7 M urea, 2 M thiourea, 4% 3-[(3-Cholamidopropyl)dimethylammonio]-1-propanesulfonate hydrate (CHAPS) *w*/*v*, 0.05 M dithiothreitol (DTT)] (Sigma-Aldrich) and processed as described above. The supernatant was labelled the hydrophobic fraction (H0 fraction). Both fractions, H1 and H0, were mixed with ten volumes of chilled acetone (Sigma-Aldrich), vortexed, and incubated at −20 °C for 12 h. The samples were centrifuged at 17,000× *g* for 30 min at 4 °C, and the supernatants were discarded. The pelleted proteins were washed with 80% acetone and centrifuged, and the supernatants were thrown away. This step was repeated twice, and the pelleted proteins were air-dried and resuspended in denaturing solution before being measured with Protein Assay Reagent (Bio-Rad, Hercules, CA, USA), with bovine serum albumin used as a standard.

### 3.3. Two-Dimensional Electrophoresis (2-DE) and Image Analysis

Two-dimensional electrophoresis was used to examine the hydrophilic and hydrophobic proteins. For the first dimension, isoelectric focusing (IF) was performed on 24 cm immobilized pH gradient (IPG) linear gradient strips of pH 5–8 (Bio-Rad) rehydrated with 1.5 mg of protein. Isoelectric focusing (IEF) was conducted at 20 °C with an Ettan IPGphor 3 IEF System (GE Healthcare, Chicago, IL, USA) at a constant 100 mA per strip under the following conditions: (1) 250 V Step and Hold for 2 h; (2) 500 V gradient until 10 Vh; (3) 2000 V gradient for 2 h; (4) 4000 V gradient for 2 h; (5) 6000 V gradient for 2 h; and (6) 8000 V Step and Hold until 100,000 Vh. After IEF, the IPG strips were incubated for 15 min in equilibration buffer (6 M urea, 30% glycerol, 2% sodium dodecyl sulphate (SDS), 0.05 M Tris-HCl pH 8.8, and 1% DTT) (Sigma-Aldrich) under gentle agitation and once again with equilibration buffer plus 2.5% (*w*/*v*) iodoacetamide (IAM) (Sigma-Aldrich) instead of DTT. In the second dimension, focused proteins were resolved in 13% polyacrylamide-SDS gels using the Ettan DALTsix Electrophoresis Unit (GE Healthcare) at 10 mA/gel for 27 h. Once the electrophoretic separation was finished, the gels were stained with colloidal Coomassie G-250 (Bio-Rad) for 12 h and unstained with ultrapure water. One gel image for each biological replicate was acquired (40 gel images in total) using the Pharos FX Plus Molecular Imager (Bio-Rad) at a 100 μm resolution and analyzed using Melanie Software v9.2 (GeneBio-SIB Swiss Institute of Bioinformatics, Lausanne, Vaud, Switzerland) for isoelectric point (*pI*), molecular mass (MW), and spot volume for densitometry determinations. The densitometric data were submitted to a one-way ANOVA, and spots were considered differentially accumulated only if they presented both a fold change ≥ 2.0 and *p* ≤ 0.001.

### 3.4. In-Gel Digestion and nLC-MS/MS Analysis

Differentially accumulated protein spots, selected based on densitometric analysis, were manually excised and de-stained before being reduced with 10 mM DTT in 25 mM ammonium bicarbonate and alkylated with 55 mM IAM. Protein digestion was carried out overnight at 37 °C using sequencing-grade trypsin (Promega, Madison, WI, USA). Nanoscale LC separation of tryptic peptides was performed with a nanoACQUITY UPLC System (Waters, Milford, MA, USA) equipped with a Symmetry C18 precolumn (5 μm, 20 mm × 180 μm, Waters) and a BEH130 C18 (1.7 μm, 100 mm × 100 μm, Waters) analytical column. The lock mass compound, [Glu1]-Fibrinopeptide B (Sigma-Aldrich), was delivered via the auxiliary pump of the nanoACQUITY UPLC System at 200 nL/min at a concentration of 100 fmol/mL to the reference sprayer of the Nano-Lock-Spray source of the mass spectrometer. Mass spectrometric analysis was carried out in a SYNAPT-HDMS Q-TOF (Waters). The spectrometer was operated in V-mode, and analyses were performed in positive ESI mode. The TOF analyzer was externally calibrated with [Glu1]-Fibrinopeptide B (Waters) from *m*/*z* 50 to 2422. The data were lock–mass corrected post-acquisition using the doubly protonated monoisotopic ion of [Glu1]-Fibrinopeptide B. The reference sprayer was sampled every 30 s. The radio frequency applied to the quadrupole was adjusted such that ions from *m*/*z* 50 to 2000 were efficiently transmitted. The MS and MS/MS spectra were acquired with MassLynx software v4.1 alternating between low-energy and elevated-energy modes of acquisition (MS^e^).

### 3.5. LC-MS/MS Protein Identification, Database Searching, and Data Analysis

MS/MS spectra data sets were used to generate PKL files with the Protein Lynx Global Server v2.4 (Waters). Proteins were then identified using PKL files and the MASCOT search engine v2.5 (Matrix Science, London, UK) against the *A. hypochondriacus* transcriptome and proteome v1.0 (23,054 sequences) available at https://phytozome.jgi.doe.gov/, accessed on 15 March 2023 [111] (in this work, the prefix “AHYPO_” and suffix “-RA” of the accession numbers in the database were removed in the main text). Trypsin was set as the specific protease, and one missed cleavage was allowed. The mass tolerance for precursor and fragment ions was set to 20 ppm and 0.1 Da, respectively. Carbamidomethyl cysteine, the oxidation of methionine, and the phosphorylation of serine, threonine, and tyrosine were specified as variable modifications. The protein identification criteria included at least two MS/MS spectra matched at a 99% level of confidence, and identifications were considered successful when significant MASCOT individual ion scores were obtained indicating identity or extensive homology statistically significant at *p* < 0.001. Identifications were considered only true for peptides with identity FDR ≤ 1%. The exponentially modified protein abundance index (emPAI) was used to estimate the relative abundance of each protein when more than one protein was identified per spot [112]. Using the corrplot package, we conducted a correlation analysis to give a visual projection of the relationship of each data set based on the protein abundances of the differential spots [113]. These abundances were also plotted as heatmaps using the ComplexHeatmaps package [114].

### 3.6. Western Blot Detection of Phosphoproteins

Hydrophobic protein fractions of the five amaranth species were resolved via SDS-PAGE and transferred to a PVDF membrane (Bio-Rad) at 15 V for 90 min using an Owl HEP-1 Semidry Electroblotter System (Thermo-Fishcer Scientific, Waltham, MA, USA). Rabbit anti-Phosphoserine/threonine/tyrosine polyclonal antibody (dilution 1:1000) (Invitrogen, Waltham, MA, USA) was used as the primary antibody and was resolved with anti-rabbit IgG antibody conjugated to alkaline phosphatase (dilution 1:30,000) (Sigma-Aldrich). The enzyme–substrate reaction was performed by incubating the membrane in a developing solution (0.1 M Tris, pH 9.5, 0.5 mM MgCl_2_, containing 0.3% p-nitroblue tetrazolium chloride and 0.15% 5-bromo-4-chloro-3-indolyl phosphate, Sigma-Aldrich). After colour development, the reaction was stopped by rinsing the membrane with distilled water.

## 4. Conclusions

Polarity-based fractionation allowed for the successful resolution of differential spots of low abundance via 2-DE and the identification of the corresponding proteins via LC-MS/MS. Amaranth GBSSI and 11S globulins had differential accumulation patterns amongst amaranth species, mainly as a consequence of variations in proteolytic processing and PTMs. Both proteins were predominantly identified in the H0 fraction; that is, they present high hydrophobicity, high aggregation, or intermolecular interactions that keep them strongly bound to the plastid membranes or protein accumulation bodies. However, the fact that they are also present in the H1 fraction, although to a lesser extent, indicates broad behaviour in terms of physicochemical properties, such as solubility, affected by surface properties that are dependent on the type or degree of modification. The comparative proteomic analyses of wild and cultivated amaranth species allowed the molecular targets related to dormancy, seed size, and stress protection to be unveiled, which could be of agrobiotechnology interest for crop improvement. Prospective work may include evaluating the effect of silencing the expression of such targets or generating overexpressing lines of these proteins through crossbreeding or recombinant DNA techniques, with those found in higher abundance in wild species being of interest, such as PARP, annexin, ω-amidase, OBAPs, and ClpB1.

## Figures and Tables

**Figure 1 plants-13-02728-f001:**
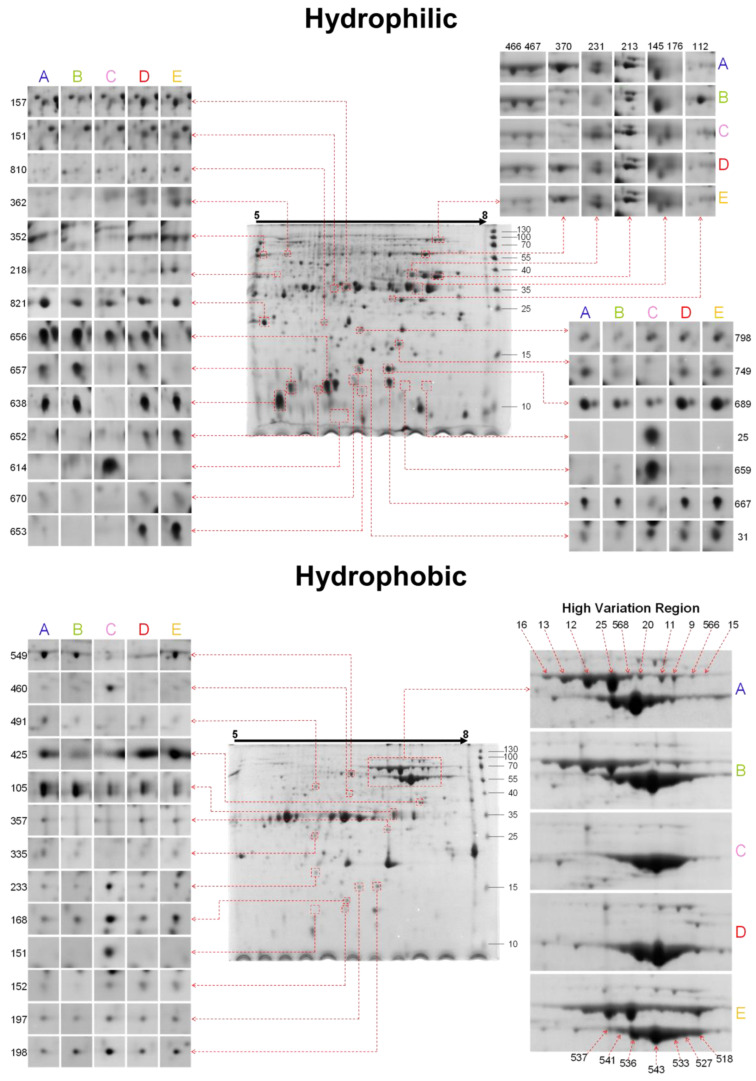
Representative 2-DE proteomic maps of the hydrophilic (H1) and hydrophobic (H0) fractions of amaranth seed species and comparative accumulation profiles of selected differential spots. A, *A. hybridus*; B, *A. powellii*; C, *A. cruentus* cv. Amaranteca; D, *A. hypochondriacus* cv. Opaca; E, *A. hypochondriacus* cv. Cristalina. The immobilized pH gradient ranges from 5 to 8 from left to right. MW markers (kDa) are indicated on the right side of the central gel images.

**Figure 2 plants-13-02728-f002:**
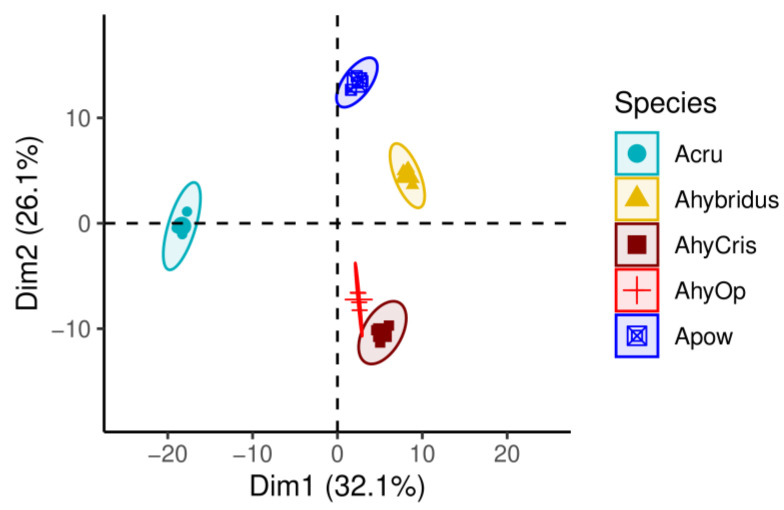
Principal Component Analysis (PCA) based on the abundances of differentially accumulated seed spots of the compared amaranth species. Ahybridus, *A. hybridus*; Apow, *A. powellii*; Acru, *A. cruentus* cv. Amaranteca; AhyOp, *A. hypochondriacus* cv. Opaca; AhyCris, *A. hypochondriacus* cv. Cristalina.

**Figure 3 plants-13-02728-f003:**
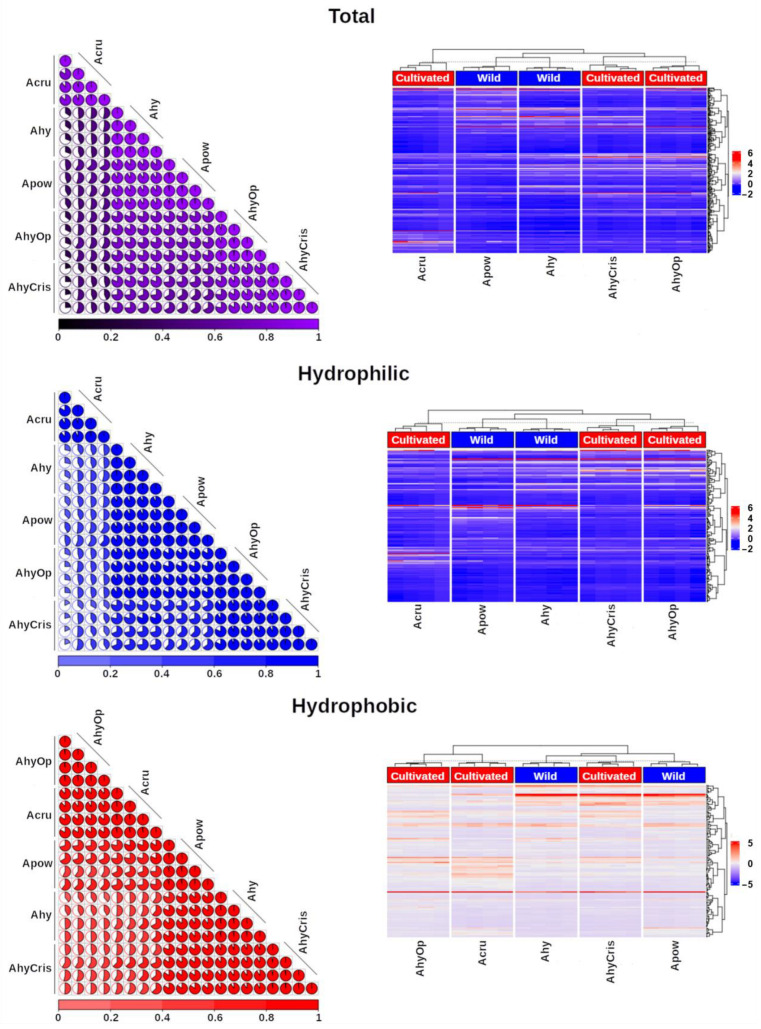
Correlograms and heatmaps showing the clustering pattern of amaranth species as a function of the protein fraction analyzed. Total protein spot abundances (up); hydrophilic (middle) or hydrophobic (bottom) protein spot abundances. Ahy, *A. hybridus*; Apow, *A. powellii*; Acru, *A. cruentus*; AhyOp, *A. hypochondriacus* cv. Opaca; AhyCris, *A. hypochondriacus* cv. Cristalina.

**Figure 4 plants-13-02728-f004:**
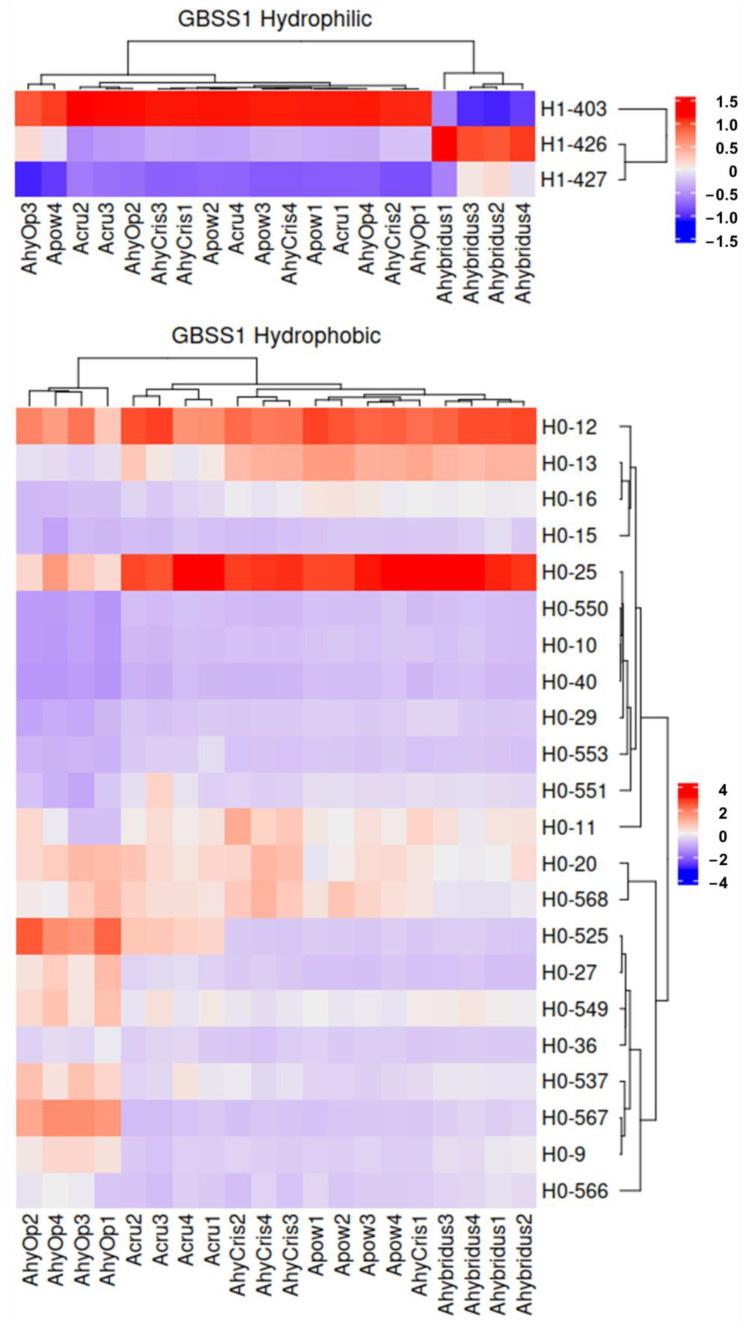
Abundance distribution of granule-bound starch synthase I (GBSSI) amongst amaranth species in hydrophilic (H1) and hydrophobic (H0) fractions. Ahybridus, *A. hybridus*; Apow, *A. powellii*; Acru, *A. cruentus*; AhyOp, *A. hypochondriacus* cv. Opaca; AhyCris, *A. hypochondriacus* cv. Cristalina.

**Figure 5 plants-13-02728-f005:**
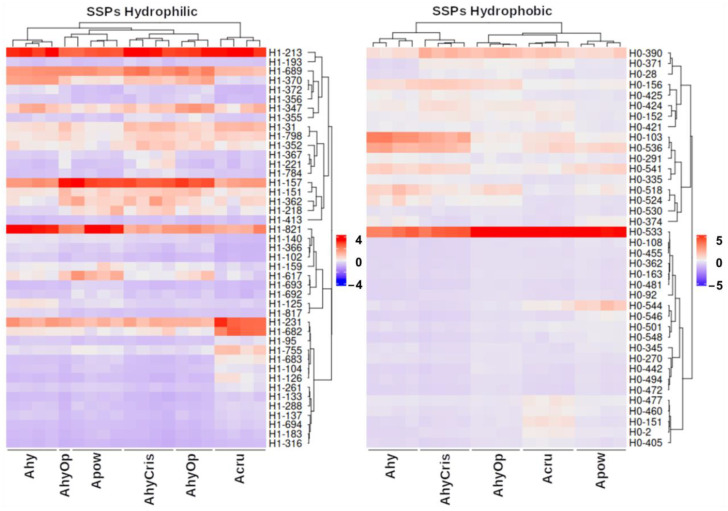
Abundance distribution of seed storage proteins (SSPs) amongst amaranth species in hydrophilic (H1) and hydrophobic (H0) fractions. Ahy, *A. hybridus*; Apow, *A. powellii*; Acru, *A. cruentus*; AhyOp, *A. hypochondriacus* cv. Opaca; AhyCris, *A. hypochondriacus* cv. Cristalina.

**Figure 6 plants-13-02728-f006:**
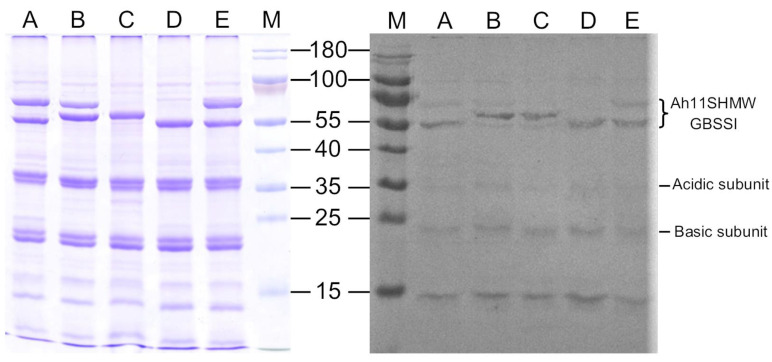
Western blot immunodetection of phosphorylated proteins in the hydrophobic protein fraction (H0) of the five amaranths. Left, electrophoretic profile of amaranths seeds hydrophobic proteins; Right, PVDF membrane after exposure to phosphoserine/threonine/tyrosine polyclonal primary antibody and reaction of alkaline phosphatase linked to an anti-rabbit IgG antibody. A, *A. hybridus*; B, *A. powellii*; C, *A. cruentus* cv. Amaranteca; D, *A. hypochondriacus* cv. Opaca; E, *A. hypochondriacus* cv. Cristalina; M, MW marker (kDa).

## Data Availability

PKL files used for LC-MS/MS identification of *Amaranthus* seed proteins derived from 2-DE spots are available at Mendeley Data DOI:10.17632/p3dhzyhvnk.2.

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
