# Peer review of "Comparative Proteomic Analysis of Wild and Cultivated Amaranth Species Seeds by 2-DE and ESI-MS/MS"

_plants, 2024, doi:10.3390/plants13192728_

Round 1

Reviewer 1 Report

Comments and Suggestions for Authors

This manuscript focuses on the comparative proteomics of mature seeds from wild and cultivated amaranth species, with an emphasis on differential phosphorylation patterns and molecular processes. Two-dimensional gel electrophoresis (2-DE) and LC-MS/MS were used to identify differentially accumulated proteins (DAPs), while Western blot immunodetection was performed to detect phosphorylated proteins. The study focuses on amaranth species, which may appeal only to a specific audience.

Major Concerns:

1. While the use of 2-DE and LC-MS/MS for studying proteomics is not novel, their application to wild and cultivated amaranth species appears to be a novel contribution. However, it seems the identification of phosphorylated proteins was based solely on molecular weight. Since LC-MS/MS is considered the gold standard for identifying proteins and phosphorylation sites, omitting this information weakens the overall strength of the study.

2. For the MS data search, phosphorylation was not included in the variable modification list. The abundance of phosphorylated peptides might be below the detection limit, but why not give it a try?

Minor:

The discussion seems a bit lengthy. 

Comments on the Quality of English Language

The manuscript is generally well-written with some minor areas where clarity and readability could be improved. 

Author Response

plants-3173106

Response to Reviewer 1 Comments             

Dear reviewer, thank you very much for taking the time to review this manuscript. Please find the detailed responses below and the corresponding corrections highlighted in track changes in the re-submitted files.

Point-by-point response to Comments and Suggestions for Authors

General Comment: This manuscript focuses on the comparative proteomics of mature seeds from wild and cultivated amaranth species, with an emphasis on differential phosphorylation patterns and molecular processes. Two-dimensional gel electrophoresis (2-DE) and LC-MS/MS were used to identify differentially accumulated proteins (DAPs), while Western blot immunodetection was performed to detect phosphorylated proteins. The study focuses on amaranth species, which may appeal only to a specific audience.

Major Comment 1: While the use of 2-DE and LC-MS/MS for studying proteomics is not novel, their application to wild and cultivated amaranth species appears to be a novel contribution. However, it seems the identification of phosphorylated proteins was based solely on molecular weight. Since LC-MS/MS is considered the gold standard for identifying proteins and phosphorylation sites, omitting this information weakens the overall strength of the study.

Major Comment 2: For the MS data search, phosphorylation was not included in the variable modification list. The abundance of phosphorylated peptides might be below the detection limit, but why not give it a try?

Response to comments: Dear reviewer we are in complete agreement with your observations. We contrast the experimental data from mass spectrometry against the amaranth database in MASCOT again, this time incorporating phosphorylation as a variable modification and we have included it in the methods, but we were not successful in detecting phosphorylated peptides. As you say, it is sometimes difficult to detect this type of modification when small amounts of protein such as those contained in a spot are analyzed. For this reason, we opted for the alternative of immunodetection for phosphorylation to present evidence to support that the "beads on a string" profiles we observed in 2-DE are because of this type of modification.

Minor Comments:

The discussion seems a bit lengthy.

Response to Minor Comments: When performing the re-analysis for identification, we established stricter parameters and thus discarded some proteins, which we eliminated from the supplementary tables and from the discussion of the text that included them, and shortened the manuscript.

Response to Comments on the Quality of English Language

Point 1: The manuscript is generally well-written with some minor areas where clarity and readability could be improved.

Response 1: Thank you very much for your comment, we have carried out a detailed review of the manuscript and tried to improve its clarity.

Reviewer 2 Report

Comments and Suggestions for Authors

The manuscript presented by Bojórquez-Velázquez et al. uses a proteomic approach based on differential protein extraction, 2DE and ESI-MS protein identification to detect differential changes on protein abundance in seeds from two wild amaranth species and three commercial varieties.

The importance of the research carried out is well presented, the experimental design and methology used is sound and carefully applied. Results are presented and highlighted in a logic, easy-to-follow and in a self-discussed manner, which is remarkable given their complexity. The pertinence and usefulness of the  2DE methodology, often considered out of date, is clearly demonstrated by the present manuscript, as the only way to show important post-translational modifications and their impact in protein biophysical characteristics and behaviour. The results obtained shed light into the complexities of seed development and protein composition of seeds, and their possible relation with seed nutritional quality as well as environmental adaptation capabilities. Therefore, I would recommend the publication of the present manuscript in the present form, subjected to minor editing corrections.

Author Response

plants-3173106

Response to Reviewer 2 Comments      

Thank you very much for taking the time to review this manuscript. Please find the detailed responses below and the corresponding corrections highlighted in track changes in the re-submitted files.

Point-by-point response to Comments and Suggestions for Authors

General Comment: The manuscript presented by Bojórquez-Velázquez et al. uses a proteomic approach based on differential protein extraction, 2DE and ESI-MS protein identification to detect differential changes on protein abundance in seeds from two wild amaranth species and three commercial varieties.

The importance of the research carried out is well presented, the experimental design and methology used is sound and carefully applied. Results are presented and highlighted in a logic, easy-to-follow and in a self-discussed manner, which is remarkable given their complexity. The pertinence and usefulness of the 2DE methodology, often considered out of date, is clearly demonstrated by the present manuscript, as the only way to show important post-translational modifications and their impact in protein biophysical characteristics and behaviour. The results obtained shed light into the complexities of seed development and protein composition of seeds, and their possible relation with seed nutritional quality as well as environmental adaptation capabilities. Therefore, I would recommend the publication of the present manuscript in the present form, subjected to minor editing corrections.

Response: Dear reviewer, thank you very much for your comments, it is nice to know that you liked our work. The manuscript has been thoroughly revised to correct any details that we have been able to detect.

Reviewer 3 Report

Comments and Suggestions for Authors

Dear Authors

you can find my comments in the attached file, best regards

Comments on the Quality of English Language

The quality of English is fine, there are occasional typos/mistakes to be revised.

Author Response

Dear reviewer, please find the response to the comments in the attached file.

Round 2

Reviewer 1 Report

Comments and Suggestions for Authors

The readability of the manuscript has improved significantly.

Reviewer 3 Report

Comments and Suggestions for Authors

Dear Authors, the current version of the paper has improved, thank you

Comments on the Quality of English Language

Minor editing of the English language